# Endoscopic Ultrasound and Intraductal Ultrasound in the Diagnosis of Biliary Tract Diseases: A Narrative Review

**DOI:** 10.3390/diagnostics14182086

**Published:** 2024-09-20

**Authors:** Akiya Nakahata, Yasunobu Yamashita, Masayuki Kitano

**Affiliations:** Second Department of Internal Medicine, Wakayama Medical University, 811-1 Kimiidera, Wakayama 641-0012, Japan; nakaaki@wakayama-med.ac.jp (A.N.); yasunobu@wakayama-med.ac.jp (Y.Y.)

**Keywords:** biliary tract diseases, endoscopic ultrasound, intraductal ultrasound

## Abstract

Endoscopic ultrasound (EUS) and intraductal ultrasound (IDUS) play very important roles in the field of biliary tract disease. Because of their excellent spatial resolution, the detection of small lesions and T- or N-staging of tumors have become possible. Additionally, contrast-enhanced EUS and the new imaging technique of detective flow imaging are reported to be useful for differential diagnosis. Furthermore, EUS-guided tissue acquisition is used not only for pathological diagnosis but also to collect tissue samples for cancer genome profiling. This review provides an overview of diagnosis utilizing the features and techniques of EUS and IDUS.

## 1. Introduction

Biliary diseases encompass a wide range of conditions, from benign diseases like choledocholithiasis to malignant ones such as biliary tract cancer, which remains a type of cancer with a poor prognosis [1,2,3]. Accurate diagnosis of these diseases is crucial because it affects patients’ outcomes [4].

Endoscopic ultrasound (EUS) and intraductal ultrasound (IDUS) have become indispensable modalities for the biliary region broadly applied in diagnosis and treatment. Because of their high spatial resolution, their diagnostic uses include diagnosis of choledocholithiasis [5,6], differential diagnosis of stricture [7,8,9] and wall thickening of the biliary tract [10,11,12], and evaluation of tumor invasion and horizontal spreading [13,14,15,16,17,18,19]. Their usefulness in diagnosing biliary tract diseases has been reported to be superior to other modalities [20,21,22]. Currently, in addition to conventional B-mode EUS, diagnostic EUS frequently makes use of techniques such as contrast-enhanced EUS (CE-EUS) [23,24] and detective flow imaging (DFI) [25]. These advances have greatly contributed to improving differential diagnostic capabilities.

EUS-guided tissue acquisition (EUS-TA) has also recently been performed for biliary tract diseases [26,27,28,29,30,31,32,33], and with the application of precision medicine to biliary tract cancers playing an increasingly important role [34,35,36], EUS-TA is becoming more important. Here, we review the current literature with respect to the roles of EUS and IDUS in the diagnosis of biliary tract diseases.

## 2. Methodology

We performed searches in PubMed and Google Scholar with the combinations of the following keywords: “biliary tract diseases”, “endoscopic ultrasound/EUS”, “intraductal ultrasound/IDUS”, “EUS-guided tissue acquisition/EUS-TA”, “EUS-guided fine needle aspiration/EUS-FNA”, “comprehensive genome profiling”, “contrast-enhanced”, “detective flow imaging/DFI”, “diagnosis”, “staging”, “biliary duct”, “gallbladder”, “ampullary”, “choledocholithiasis”, “carcinoma”, “biliary stricture”, “wall thickening”, “polyp”, and “maljunction”. We selected original articles and reviews written in English. They were full papers published up to August 2024. All the images, photos, and schemes used in this review are original. The ultrasound images obtained through EUS and IDUS were taken from our hospital’s examination database, with personal information removed. Prior consent was obtained from the patients to the use of these images in research. The photos of the devices were taken by the authors, and the schemes were created by us.

## 3. Endoscopic Ultrasound (EUS)

EUS uses an endoscope equipped with an ultrasound transducer at its tip. The frequency typically ranges from 5 to 20 MHz, allowing for switching according to the lesion. EUS is usually used to observe lesions of the digestive tract, the biliary tract, the pancreas, parts of the liver, and lymph nodes in the mediastinum and the abdominal cavity [37]. One of the important features of EUS is its high spatial resolution, which is achieved through close-up observation of the target organs. Whereas transabdominal ultrasound has limitations in its observations of the pancreatobiliary system due to gastrointestinal gas, thick subcutaneous fat, and bone, EUS enables visualization of the organs from the gastrointestinal tract, such as the stomach and duodenum, and is less affected by such intervening tissues. There are two types of EUS: the radial type, in which ultrasound images are acquired perpendicular to the endoscope, and the convex type (linear type), in which ultrasound images are acquired parallel to the endoscope. The former can depict a 360° field of view, making anatomical orientation easier. By contrast, the convex type has a 180° field of view, although it is useful for EUS-TA and interventional EUS (Figure 1).

## 4. Intraductal Ultrasound (IDUS)

Intraductal ultrasound (IDUS) uses a thin ultrasound probe (with a diameter of 2–3 mm) with a frequency ranging from 15 to 30 MHz that can be inserted through the endoscope channel. The probe is sufficiently small to allow its insertion into the bile duct or the pancreatic duct, in which it can obtain ultrasound images. The ultrasound probe has a high spatial resolution, allowing for visualization of the positional relationships between surrounding blood vessels, the bile duct, the pancreatic duct, and the sphincter of Oddi at the major papillae [38] (Figure 2).

## 5. Contrast-Enhanced EUS

The contrast agents used in ultrasound have a structure consisting of microbubbles covered with carbohydrates or phospholipids. Commercial intravenous ultrasound contrast agents started with Levovist^®^ (Bayer Schering Pharma, Berlin, Germany), which is now referred to as a first-generation contrast agent, and have now developed into second-generation contrast agents, such as Sonazoid^®^ (GE healthcare, Tokyo, Japan), SonoVue^®^ (Bracco, Milan, Italy), and Definity^®^ (BristolMyers Squibb Medical Imaging, New York, NY, USA), which make contrast-enhanced EUS (CE-EUS) possible even under a low acoustic power [39]. These agents enable stable imaging and prolonged observation with contrast harmonic imaging, allowing for real-time assessment of micro-scale blood flow. They also have a low risk of allergic reactions and are suitable for patients with renal dysfunction because the gas contained in the microbubbles is excreted through exhalation [40]. They can also be used in those who are allergic to iodine-based contrast agents. It is also possible and economically viable to use them for repeat examinations.

## 6. EUS-Guided Tissue Acquisition

EUS-guided tissue acquisition (EUS-TA) is a procedure for obtaining tissue from lesions in the submucosa or located outside the gastrointestinal tract. Convex-type EUS is used to visualize the target lesion, and a specialized needle is inserted through the scope channel to puncture the lesion and obtain tissue samples. Since its utility was first reported by Vilmann et al. [41], EUS-TA has been adopted widely. Nowadays, it is the reference standard for diagnosis and is useful for tumor staging and for deciding on a treatment strategy. The needles used for puncture are classified as aspiration needles (FNA needles) or biopsy needles (FNB needles) according to their shape and size (typically 19 G, 22 G, or 25 G). FNB needles are superior to FNA needles for collecting large tissue samples [42,43], which are useful for performing immunohistochemistry staining and comprehensive genome profiling [44]. However, FNA needles offer a superior puncture performance [45] and allow samples to be obtained from smaller lesions [46]. It is important to choose the appropriate needle according to the specific purpose.

## 7. Diagnosis of Biliary Duct Diseases

### 7.1. Diagnosis of Choledocholithiasis

Choledocholithiasis is the most common benign biliary disease encountered in clinical practice and can sometimes be complicated by severe acute cholangitis, which requires biliary drainage. The diagnostic modalities for choledocholithiasis include abdominal ultrasound (AUS), computed tomography (CT), magnetic resonance cholangiopancreatography (MRCP), and EUS. In some cases, IDUS may also be employed. The sensitivity for detecting choledocholithiasis is 63% for AUS, 71% for CT, and 96% for EUS [47]. AUS and CT tend to show a particularly poor diagnostic ability for choledocholithiasis in patients with small stones or stones in a non-dilated common bile duct. The specificity of EUS for detecting choledocholithiasis is near to 100% and is higher than that of AUS (95%) and CT (97%) [47]. MRCP is also an effective modality for the diagnosis of choledocholithiasis; one meta-analysis found a sensitivity and specificity for diagnosing choledocholithiasis of 96% and 92%, respectively, for EUS and 85% and 90% for MRCP [5]. Suzuki et al. [6] also reported that EUS had a superior diagnostic ability to MRCP for diagnosing choledocholithiasis that was missed on CT, with the high spatial resolution of EUS making it the most reliable and efficient diagnostic modality for small lesions [6,47,48,49,50,51,52] (Figure 3).

IDUS is also known to be effective for detecting choledocholithiasis; Linghu et al. [53] reported that its accuracy and sensitivity in the diagnosis of extrahepatic bile duct stones were both 100%. Another study reported [54] that IDUS revealed residual stones in 38% of cases despite normal cholangiography. Whether IDUS should be performed to confirm the presence of residual stones as a routine procedure is not clearly defined, and further investigation is needed.

In conclusion, if the presence of stones cannot be ruled out by other modalities, it is advisable to actively perform EUS or IDUS.

### 7.2. Differential Diagnosis of Biliary Stricture

Biliary stricture is a commonly encountered condition, and distinguishing between benign and malignant lesions is crucial. Benign diseases include primary sclerosing cholangitis (PSC), IgG4-related sclerosing cholangitis (IgG4-SC), inflammatory stricture, postoperative stricture, and other secondary cholangitis, and their differentiation from biliary tract cancer is very important. A previous report showed that 8–43% of biliary strictures suspected to be malignant and subsequently resected were actually benign [55]. Therefore, since surgery for biliary tract cancer is highly invasive, an accurate diagnosis is important to avoid unnecessary procedures. It is also important to diagnose IgG4-SC because it is curable if the appropriate treatment with steroids is provided.

Previous studies have reported that EUS and IDUS were effective for differential diagnosis between IgG4-SC, PSC, and cholangiocarcinoma [7,8,9]. Table 1 lists the typical findings and features of these diseases. A key point in differential diagnosis concerns wall thickness, which is observed in regions of non-stricture in IgG4-SC, and this sign is important for a differential diagnosis from cholangiocarcinoma (with sensitivity of 95%, specificity of 91%, and accuracy of 94%) [7]. Recently, the use of immune checkpoint inhibitors has increased, leading to more reports of cholangitis caused by immune-related adverse events [56,57,58,59]. Differentiating a diagnosis of cholangiocarcinoma from IgG4-SC and PSC is necessary, and while it is reported that diffuse bile duct wall thickness is characteristic, it tends not to involve bile duct strictures, unlike in PSC and IgG4-SC. However, there are still few reported cases of this, and further accumulation of cases is needed.

Recently, the effectiveness and safety of EUS-TA for biliary tract tumors have been frequently reported. Table 2 shows several results concerning EUS-TA of malignant biliary strictures [26,27,28,29]. These results were validated in many cases, with EUS-TA demonstrating a high diagnostic capability and few adverse events. In systematic reviews comparing the use of ERCP and EUS-TA for the diagnosis of malignant biliary stricture, the sensitivities of ERCP and EUS-TA for tissue diagnosis of malignant biliary stricture were 49% and 75%, respectively; their specificities were 96% and 100%; and the accuracy rates were 61% and 79% [27]. Jo et al. [28] compared the diagnostic performance of EUS-TA- and ERCP-based tissue sampling for malignant biliary obstruction and revealed overall diagnostic sensitivity and accuracy rates of 74% and 76%, respectively, for EUS-TA; 57% and 61% for ERCP; and 86% and 87% for a combination of EUS and ERCP. However, the risk of peritonitis due to bile leakage and potential tumor seeding should be heeded [60]. We conclude that in cases where a diagnosis cannot be obtained with ERCP, performing EUS-TA for biliary stricture should be considered with careful consideration of the risks and benefits.

### 7.3. Diagnosis of Cholangiocarcinoma

As mentioned above, cholangiocarcinoma is the most important disease in biliary stricture, and it requires an exact diagnosis. EUS and IDUS have been reported to be effective in the qualitative diagnosis of cholangiocarcinoma, including duct wall invasion, intraductal progression, and invasion into the surrounding organs and vessels.

In qualitative diagnosis by EUS, the appearance of a hypoechoic mass completely occluding the lumen and a heterogeneously increased irregular wall thickness in the distal bile duct were found to be highly predictive of and sensitive for detecting malignancy originating from the distal bile duct, with a reported sensitivity of 75.8% and 68.1%, respectively [61]. In comparison, the sensitivity, specificity, and accuracy rates for a qualitative diagnosis of bile duct strictures (wall thickening, irregular margins, etc.) by IDUS are high, at 93.2%, 89.5%, and 91.4%, respectively [62].

#### 7.3.1. T-Staging of Cholangiocarcinoma

EUS and IDUS can clearly delineate the structure of the bile duct wall. The normal bile duct wall structure consists of two to three layers, with the inner layer being hypoechoic and the outer layer being hyperechoic (Figure 4). The inner hypoechoic layer reflects the mucosal layer, the fibromuscular layer, and part of the subserosal layer. The outer hyperechoic layer corresponds to the subserosa and the serosa. If this lateral hyperechoic layer is irregular or interrupted, it should be considered suspicious for invasion.

In the diagnosis of extrahepatic cholangiocarcinoma progression using EUS, the accuracy rates for diagnosis of invasion into the surrounding tissue, invasion into the pancreatic parenchyma, and invasion into the portal vein were reported to be 67–83%, 70–83%, and 80–92%, respectively [63,64]. Although these values indicate EUS has a high diagnostic performance, there is a weak point in that its diagnostic ability for vascular invasion “in the hilar” is inferior to that for vascular invasion in the distal bile duct [65].

The accuracy rates of IDUS for the diagnosis of invasion into the right hepatic artery, the pancreatic parenchyma, and the portal vein are 100%, 93%, and 93%, respectively [13]. IDUS is reported to have higher accuracy rates for diagnosis and T-staging than EUS (IDUS: 77.7% vs. EUS: 54.1%) [14].

CE-EUS is useful for the T-staging of biliary tract cancer. Otsuka et al. [15] reported that CE-EUS is superior to contrast-enhanced CT and conventional EUS in the detection of invasion beyond the bile duct wall. Imazu et al. [16] also reported that CE-EUS showed a higher diagnostic accuracy for the depth of invasion of biliary tract cancer than conventional B-mode EUS (accuracy rates: 92.4% vs. 69.2%; *p* < 0.05). Diagnosis of the longitudinal tumor extent is especially important in hilar cholangiocarcinoma because the operative methods differ depending on the tumor extent, and preoperative misdiagnosis of the tumor extent makes R0 resection impossible. The reported accuracy rates for the longitudinal extent of hilar cholangiocarcinoma on IDUS range from 85% to 90% [17,18].

#### 7.3.2. N-Staging of Cholangiocarcinoma

The overall survival of lymph node metastasis-negative patients with cholangiocarcinoma is better than that of lymph node metastasis-positive patients [66]. The detection of at least one malignant lymph node metastasis is associated with a lower median survival and mortality [67]. Therefore, it is important to detect any lymph node metastases (with N-staging). EUS is superior to cross-sectional imaging such as CT and MRI in the detection of lymph nodes (86% vs. 47%; *p* < 0.001) [67]. Miyata et al. reported that EUS findings are useful for distinguishing between benign and malignant lymph nodes, with the findings suggesting malignancy including a long axis ≥ 20 mm, a round shape, a sharp edge, hypoechogenicity, the absence of central intranodal blood vessels, and heterogeneity on contrast enhancement [68].

## 8. Diagnosis of Gallbladder Diseases

### 8.1. Differential Diagnosis of Wall Thickening Lesions

The gallbladder wall consists of two layers: the inner layer includes the mucosa, the muscularis propria, and part of the subserosa, which appear hypoechoic, and the outer layer consists of part of the subserosa and the serosa, which appear hyperechoic. EUS can usually depict these two layers [69] (Figure 4).

Wall thickening of the gallbladder is defined as a thickened wall measuring more than 3 mm in diameter [70]. Its differential diagnoses are varied, ranging from benign diseases such as chronic cholecystitis, xanthogranulomatous cholecystitis, and adenomyomatosis (ADM) to malignant diseases such as gallbladder carcinoma.

Characteristic EUS findings of ADM include comet-tail artifacts within the thickened wall and cystic anechoic spots, which are crucial for diagnosis [10] (Figure 5). EUS can depict these findings in detail, whereas gallbladder carcinoma usually does not have such echoic findings. Findings suggestive of malignancy include wall thickening of 10 mm or more, heterogeneous internal echogenicity with regions of low echogenicity, and loss of the layer structure [11] (Table 3). However, it is necessary to carefully observe whether there are any irregularities on the surface of ADM because carcinoma can coexist with ADM [71].

Xanthogranulomatous cholecystitis is a subtype of cholecystitis. Its characteristics include xanthoma cells with bile pigments forming granulomas in the gallbladder wall and inflammatory infiltration extending to the surrounding organs as if cancer were invading; however, it is difficult to distinguish it from gallbladder carcinoma. The characteristic ultrasound findings of xanthogranulomatous cholecystitis are reported to include a lack of wall disruption and intramural hypoechoic nodules [12], but clinicians often face challenges in its differential diagnosis. It may be difficult to distinguish it from malignancy by EUS alone.

### 8.2. Differential Diagnosis of Protuberant Lesions

Protuberant lesions of the gallbladder encompass a wide range of non-neoplastic conditions such as cholesterol polyps, hyperplastic polyps, adenomyomatosis, and inflammatory polyps, as well as neoplastic conditions, including adenomas and gallbladder carcinoma. Given this broad spectrum, differential diagnosis is crucial. EUS plays an important role in this differentiation, with it being reported to have a high diagnostic ability for protuberant gallbladder lesions.

Azuma et al. [72] reported that EUS had a high differential diagnostic ability for gallbladder lesions (less than 20 mm) and that it was superior to AUS. Their study showed sensitivity, specificity, and positive and negative predictive values for EUS and AUS in the diagnosis of malignancy of 91.7% vs. 54.2%, 87.7% vs. 53.8%, 75.9% vs. 54.2%, and 96.6% vs. 94.6%, respectively. However, for lesions smaller than 10 mm, the differentiation of malignancy by EUS can be challenging [73]. In the European guidelines, cholecystectomy is recommended for gallbladder polyps larger than 10 mm. However, even if the polyps are smaller than 10 mm, they can still be neoplastic lesions [74,75], and we must be careful.

It was reported that polyps larger than 14 mm are useful for distinguishing between benign and malignant lesions [76]. While it is easy to suggest that the larger the polyp, the higher the proportion of malignancy, there are also scoring systems that are useful for distinguishing between smaller polyps.

An EUS scoring system for diagnosing whether gallbladder polypoid lesions are benign or malignant has been suggested, with this system including features such as the layer structure, echo patterns, margin of the polyp, stalk, and number of polyps; polyps with a score of six or greater are considered to be at high risk of malignancy (Table 4) [77]. This scoring system has utility for distinguishing whether gallbladder lesions (with a size of 5–15 mm) are benign or malignant. Findings suggestive of malignancy include loss of the layer structure, an isoechoic heterogeneous pattern, lobulated margins, sessile type, and the presence of multiple polyps [77].

Contrast-enhanced EUS (CE-EUS) is also useful for discriminating benign from malignant gallbladder lesions. The evaluation of tumor vascularity aids in distinguishing between benign and malignant lesions. Choi et al. [78] reported that an irregular vessel pattern in a gallbladder polyp on CE-EUS could diagnose malignancy with a sensitivity and a specificity of 90.3% and 96.6%, respectively, and the presence of perfusion defects in a gallbladder polyp determined by CE-EUS could diagnose malignancy with a sensitivity and specificity of 90.3% and 94.9%, respectively. In definitely determining a diagnosis of gallbladder carcinoma, the sensitivity and a specificity of CE-EUS were 93.5% and 93.2%, respectively, being slightly superior to conventional EUS (90.0% and 91.1%).

Yamashita et al. [79] reported that a novel technique called detective flow imaging (DFI), which can visualize fine vessels and slow flow not detectable with conventional color Doppler or power Doppler techniques, was useful for distinguishing between benign and malignant gallbladder lesions. This technique does not require contrast agents or additional examination time and can be evaluated more easily. In the diagnosis of gallbladder carcinoma, the sensitivity, specificity, and accuracy of detecting irregular vessels with DFI-EUS were 89%, 100%, and 92%, respectively [79].

Finally, we believe that utilizing an EUS scoring system and understanding vascular images through CE-EUS or DFI may be useful in distinguishing malignancy in smaller lesions.

EUS-TA is reported to be effective for pathological examination of gallbladder lesions. Giri et al. reported a systematic review and meta-analysis on the utility and safety of EUS-TA for gallbladder lesions [30]. This showed that the sensitivity, specificity, and accuracy for the diagnosis of malignant lesions were 90%, 100%, and 94.1%, respectively. Adverse events showed a pooled incidence of 1.8%, but none of the patients had a serious adverse event. However, as when EUS-TA is used for biliary tract cancer, we cannot deny the risk of peritonitis due to bile leakage and potential tumor seeding.

In other words, in operable cases, a first tissue sampling via ERCP is recommended. If a diagnosis is not established with ERCP, EUS-TA for lymph node metastases or liver metastases may be considered. For unresectable cases, it is preferable to consider EUS-TA from locations where tissue sampling can reliably be performed.

The number of chemotherapy drugs available for biliary tract cancers, including gallbladder carcinoma, is rather limited. Recently, comprehensive genome profiling (CGP) has attracted attention in respect to devising treatment strategies, and EUS-TA plays an important role not only in pathological diagnosis but also in the field of CGP [34,35]. In a meta-analysis by Yoon et al., the diagnostic sensitivity for biliary tract cancers was 67% with biopsy and 73.6% with EUS-TA [80]. The sensitivity of transpapillary biopsy is not so high, which may be challenging for CGP. Yanaidani et al. reported the utility of EUS-TA for CGP of biliary tract cancers [36]. They suggested that an FNB needle of 22 G or larger should be used because it allows suitable specimens for CGP to be obtained, with such specimens having accuracy comparable to that of surgical specimens.

### 8.3. Diagnosis of Gallbladder Carcinoma

In gallbladder carcinoma, EUS is useful not only for qualitative diagnosis but also for staging and pathological diagnosis [81,82,83]. It is important to evaluate the depth of gallbladder wall invasion and the extent of invasion into the surrounding organs because these are involved in determining the treatment plan.

#### 8.3.1. T-Staging of Gallbladder Carcinoma

As mentioned above, the gallbladder wall is depicted as two layers. Because the outer layer consists of part of the subserosa and the serosa, thinning and disruption of the outer layer indicates suspicion of tumor invasion deeper than the subserosa. Sugimoto et al. reported that EUS can diagnose subserosal invasion of gallbladder carcinoma by focusing on the condition of the outer layer (with sensitivity of 97.1%, specificity of 86.7%, and accuracy of 93.8%) [19,84]. CE-EUS is reported to be useful for diagnosis of tumor invasion because it allows clear visualization of the layer structure [16].

#### 8.3.2. N-Staging of Gallbladder Carcinoma

The presence or absence of lymph node metastasis in gallbladder cancer affects the surgical approach and patient prognosis, similar to the T-stage. Therefore, the detection of lymph node metastasis is very important. EUS is also useful for diagnosing the N-stage of gallbladder cancer [85]. Mitake et al. reported that the sensitivity, specificity, and accuracy of EUS for detecting regional lymph node metastasis were 81.8%, 92.9%, and 89.7%, respectively [86].

## 9. Diagnosis of Other Diseases

### 9.1. Diagnosis of Pancreaticobiliary Maljunction

Pancreaticobiliary maljunction is a congenital anomaly where the pancreatic and bile ducts join outside the duodenal wall. This condition leads to reflux of pancreatic juice and bile, which significantly increases the risk of bile duct gallbladder cancer and pancreatitis. Diagnosis can be confirmed by observing the absence of the sphincter of Oddi’s influence at the junction of the pancreatic and bile ducts.

EUS is an effective modality for diagnosing pancreaticobiliary maljunction. It allows for confirmation that the bile duct and pancreatic duct converge within the pancreatic parenchyma. The diagnostic accuracy of EUS for pancreaticobiliary maljunction is reported to be high, ranging from 88% to 100% [87,88,89,90] (Figure 6). As a diagnostic method for pancreaticobiliary maljunction, direct cholangiography via ERCP is also useful. Additionally, measurement of the amylase levels in bile is possible. IDUS is performed following ERCP, confirming the junction of the pancreatic and bile ducts outside the duodenal wall. However, these methods are invasive, and post-ERCP pancreatitis can be a concern.

### 9.2. Diagnosis of Ampullary Tumors

Ampullary tumors can be classified into epithelial and non-epithelial types. Epithelial tumors include adenoma and adenocarcinoma. Non-epithelial tumors include neuroendocrine tumors, among others. A diagnosis can often be achieved through conventional endoscopic biopsies; therefore, reports on EUS-TA for ampullary tumors are few (Table 5) [31,32,33]. EUS-TA is expected to be useful for diagnosing ampullary tumors that cannot be diagnosed through conventional pathological examinations, such as biopsies taken from within the papilla and/or brush cytology during ERCP. The treatment for ampullary tumors traditionally involved pancreaticoduodenectomy, which is highly invasive and associated with high mortality rates and high postoperative adverse event rates [91]. Therefore, the less invasive approach of endoscopic papillectomy has gained attention for ampullary adenoma and adenocarcinoma which do not invade the pancreatic or bile duct. Endoscopic papillectomy for ampullary adenoma has obtained a consensus in guidelines [92], but for adenocarcinoma, invasion up to the T1a depth is acceptable. Evaluation of factors such as the tumor invasion beyond the Oddi sphincter and involvement of the pancreatic or bile duct is crucial for treatment decisions. Therefore, EUS and IDUS are essential modalities for assessing the depth of ampullary tumors.

#### 9.2.1. T-Staging of Ampullary Tumors

A meta-analysis on the diagnostic accuracy of EUS for local staging of ampullary tumors reported sensitivity and specificity for T-staging of 89% and 87%, respectively, for T1; 76% and 91% for T2; 81% and 94% for T3; and 72% and 98% for T4 [93]. By comparison, the sensitivity and specificity of IDUS for T-staging were reported to be 99% and 88%, respectively, for T1; 73% and 91% for T2; and 79% and 97% for T3.

#### 9.2.2. N-Staging of Ampullary Tumors

A meta-analysis on the diagnostic accuracy of EUS and IDUS for N-staging of ampullary tumors reported the sensitivity and specificity as 61% and 77% and 61% and 92%, respectively [93].

## 10. Conclusions

EUS and IDUS perform important roles in the diagnosis of biliary diseases. It is evident that the remarkable spatial resolution of EUS and IDUS is their most noteworthy feature. These modalities can detect choledocholithiasis with high sensitivity, even when other modalities cannot, and they contribute significantly to cancer staging. Additionally, the use of contrast-enhanced modes and DFI further enhances their diagnostic capabilities. The utility of EUS-TA for tissue diagnosis has also been widely reported, making these modalities indispensable in biliary tract diseases. They are expected to continue advancing in the future.

## Figures and Tables

**Figure 1 diagnostics-14-02086-f001:**
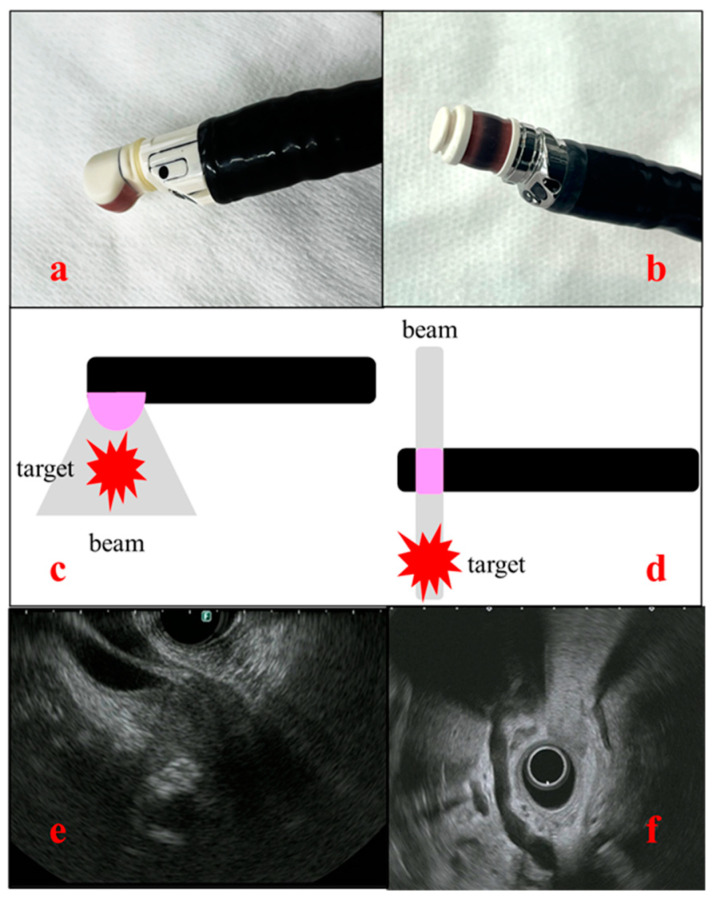
Images and schema of the two types of endoscopic ultrasound (EUS). (**a**) Convex-type EUS (GF-UCT260, Olympus, Tokyo, Japan); (**b**) radial-type EUS (GF-UE290, Olympus, Japan); (**c**) scheme of convex-type EUS; (**d**) scheme of radial type EUS; (**e**) ultrasound view of convex-type EUS; and (**f**) ultrasound view of radial-type EUS.

**Figure 2 diagnostics-14-02086-f002:**
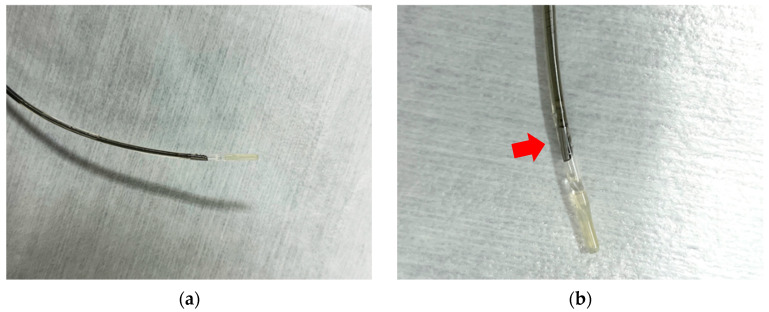
These are images of intraductal ultrasound probes: (**a**,**b**) the ultrasound probe is attached at the point indicated by the red arrow (UM-DG20-31R, Olympus, Japan).

**Figure 3 diagnostics-14-02086-f003:**
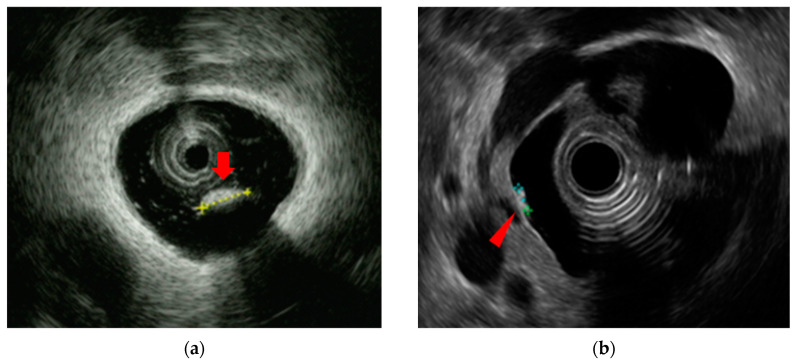
Detection of small choledocholithiasis by intraductal ultrasound (IDUS) and endoscopic ultrasound (EUS): (**a**) IDUS image of choledocholithiasis (arrowhead, 5 mm); (**b**) EUS image of choledocholithiasis (arrowhead, 5 mm).

**Figure 4 diagnostics-14-02086-f004:**
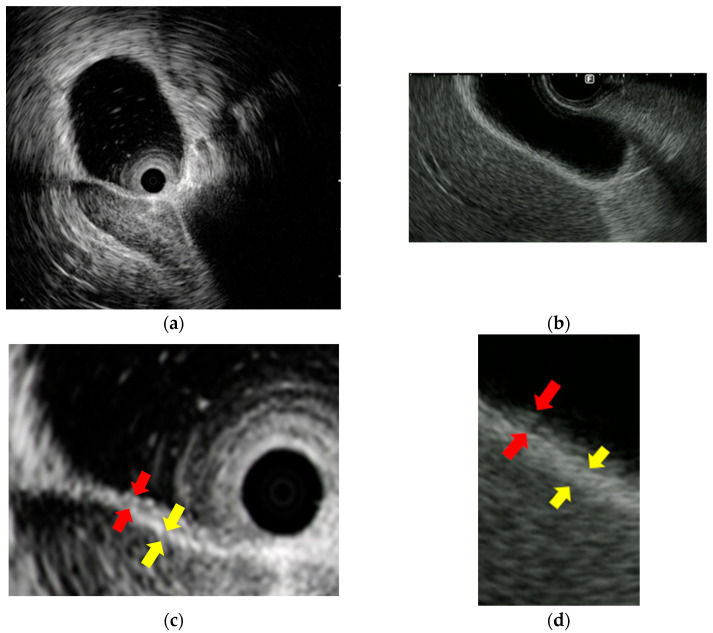
Images of the normal structure of the biliary duct wall and gallbladder wall: (**a**,**b**) images of the biliary duct wall on intraductal ultrasound and the gallbladder wall on endoscopic ultrasound; (**c**,**d**) red arrows show the inner hypoechoic layer corresponding to the mucosa, the muscularis propria, and part of the subserosa. Yellow arrows show the outer hyperechoic layer corresponding to part of the subserosa and the serosa.

**Figure 5 diagnostics-14-02086-f005:**
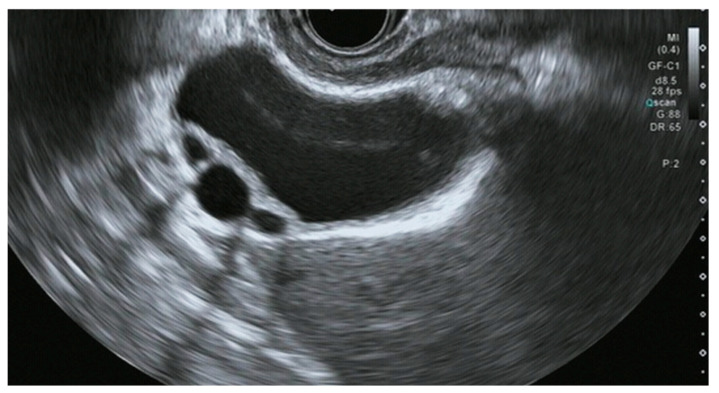
Endoscopic ultrasound image of adenomyomatosis: thickened wall and cystic anechoic spots are visible. The cystic spots show Rokitansky–Aschoff sinuses.

**Figure 6 diagnostics-14-02086-f006:**
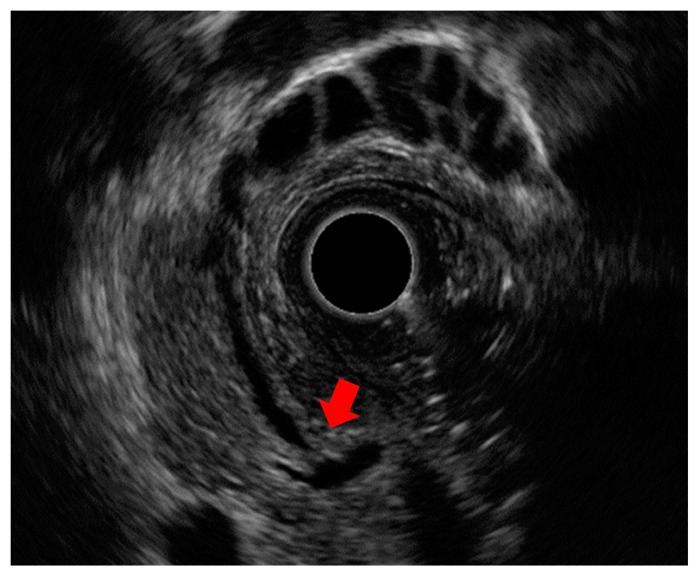
Pancreaticobiliary maljunction: an endoscopic ultrasound image showing the pancreatic duct and the bile duct converging outside the duodenal wall.

**Table 1 diagnostics-14-02086-t001:** Differences in IDUS findings of biliary stricture between IgG4-related sclerosing cholangitis (IgG4-SC), primary sclerosing cholangitis (PSC), and cholangiocarcinoma [7,8,9].

	IgG4-SC	PSC	Cholangiocarcinoma
Wall thickness	circular–symmetric	circular–asymmetric	asymmetric
Three-layer structure	preservation	disappearance	disappearance
Internal echo	homogeneous	heterogeneous	heterogeneous
Inner margin	smooth	irregular	irregular
Outer margin	smooth	unclear	irregular, interruption

**Table 2 diagnostics-14-02086-t002:** EUS-TA for malignant biliary stricture.

Author	Number of Patients	Sensitivity (%)	Specificity (%)	Accuracy (%)	Adverse Events (%)
Anahita Sadeghi et al. [26]	957	80	97	93	1 (severe: 0.3%)
De Moura et al. [27]	294	75	100	79	-
Jo et al. [28]	263	73.6	100	76.1	0.7
Praveen Mathew et al. [29]	77	91	100	93	-

**Table 3 diagnostics-14-02086-t003:** EUS gallbladder wall findings suggestive of malignancy.

	Benign	Malignant
wall thickening	<10 mm	≥10 mm
hypoechoic internal echogenicity	absent	present
internal echo pattern	homogeneous	heterogenous
wall layer	present	disrupted

**Table 4 diagnostics-14-02086-t004:** Scoring system for EUS findings on gallbladder polypoid lesions (presented with some modifications).

	Score
Layer structure	
preserved	0
lost	6
Echo pattern	
hyperechoic spots	0
hyperechoic homogeneous	1
isoechoic homogeneous	2
isoechoic heterogeneous	5
Margin of the polyp	
not lobulated	0
lobulated	4
Stalk	
pedunculated	0
sessile	3
Number of polyps	
multiple	0
single	2

**Table 5 diagnostics-14-02086-t005:** EUS-TA for ampullary tumors.

Author	Number of Patients	Sensitivity (%)	Specificity (%)	Accuracy (%)	Complications
Defrain C et al. [31]	35	82	100	89	-
Chang et al. [32]	20	-	-	35	-
Ogura et al. [33]	10	100	100	100	0

## Data Availability

The data supporting the reported results can be found in the references at the end of this article.

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
