# Peer review of "Endoscopic Ultrasound and Intraductal Ultrasound in the Diagnosis of Biliary Tract Diseases: A Narrative Review"

_diagnostics, 2024, doi:10.3390/diagnostics14182086_

Round 1
Reviewer 1 Report
Comments and Suggestions for Authors
I read with interest this narrative review on the role of EUS and IDUS in the management of biliary diseases.
The paper is a comprehensive and quite clear examination of the role of EUS and IDUS in the management of biliary tract diseases; I have no major revisions to suggest.
Minor revisions suggested:
- In the chapter regarding the role of EUS in biliary drainage, Authors should consider to mention the introduction of LAMS, which have revolutionized operative EUS, making EUS-guided biliary drainage faster and promoting its widespread adoption.
- In paragraph 7.2, page 10, there is a typo (layer stricture instead of layer structure).
A linguistic review could enhance the readability of the paper
Author Response
Dear Editors and Reviewers
Thank you very much for reviewing our manuscript and offering valuable advice.
We have addressed your comments with point-by-point responses, and revised the manuscript accordingly.
Responses to Reviewer 1
We appreciate the reviewer’s constructive comments.
Comments 1:In the chapter regarding the role of EUS in biliary drainage, Authors should consider to mention the introduction of LAMS, which have revolutionized operative EUS, making EUS-guided biliary drainage faster and promoting its widespread adoption.
Response 1:Thank you for your helpful recommendation. As you mentioned, we believe LAMS have revolutionized interventional EUS and should mention the introduction of LAMS too. However, we have decided to omit the section of the treatment. Because this review was pointed out that the section of the treatment was explained insufficiently by other reviewer, we have decided not to address treatments. The contents of diagnosis have already been occupied by more than 4000 words. Because adding more contents of treatment would make the content too extensive, we believe that this review should be focused on only diagnosis.
Comments 2:In paragraph 7.2, page 10, there is a typo (layer stricture instead of layer structure).
Response 2:Thank you for your comment. “Layer structure” is correct, so we have revised this.
Reviewer 2 Report
Comments and Suggestions for Authors
Thank the Editors for the opportunity to review this paper. It is a review about the endoscopic and intraductal ultrasound in the diagnosis and treatment of biliary tract diseases. The article describes the methods used to detect the mentioned diseases and despite the extensive description, I noticed a few things that should be expanded.
Introduction: it is a very brief introduction to the topic. First of all, there is a lack of literature. The introduction also requires an explanation of why the authors decided to take up the topic of the work.
Materials and methods: it is completely unknown what methods the authors used in the review, and what type of review the paper constitutes.
Chapters 2-5: despite the authors' broad knowledge of the presented methods, we do not have any references. If the reader wanted to delve deeper into the subject of a given method, he would not know where to look for the information that the authors provided.
There is no information provided as to where the figures come from, whether they are from literature or private photos of the authors.
Chapters 6-8: First 4 subsections describes the diseases, the last one is a explanation of the method. I don't really understand why the authors divided the section in this way. Moreover, in the flood of information it is hard to find a conclusion from the paper. The review should provide conclusions for the reader, I don't notice it in this work.
Chapter 9: again we don't know where the figure comes from. Comparisons of methods refer only to selected diseases, which is understandable, because otherwise it would go far beyond the scope of a single publication. However, the authors have described the previous chapters in great detail, which leaves some unsaid information. It is worth considering whether the work should not focus on a smaller scope, but described in a more precise manner. A positive aspect of the work is the tables, which make the analysis more readable, but they are missing in all comparisons.
Conclusions: does not contain any valuable conclusions that the reader would expect.
In summary, the work is interesting, it describes the topic very broadly. It seems that the authors spent a lot of time searching the literature. However, it lacks a plan for presenting the problem, a summary of individual methods, and possible tips. Including editorial errors, I believe that publication after major revisions can be considered.
Author Response
Dear Editors and Reviewers
Thank you very much for reviewing our manuscript and offering valuable advice.
We have addressed your comments with point-by-point responses, and revised the manuscript accordingly.
Responses to Reviewer 2
We appreciate the reviewer’s constructive comments.
Comments 1:Introduction: it is a very brief introduction to the topic. First of all, there is a lack of literature. The introduction also requires an explanation of why the authors decided to take up the topic of the work.
Response 1:Thank you very much for your important comments. As you mentioned, because there is a lack of literature, we have added the more explanation to the introduction. We shaded the edited section in yellow and added strikethrough to sentence we would delete. (page1 line24-27, line33-34, line 41-45, line46)
Comments 2:Materials and methods: it is completely unknown what methods the authors used in the review, and what type of review the paper constitutes.
Response 2:We apologize for my insufficient explanation. We have added that this review is a narrative review in the title of manuscript. (Page1 line3)
Comments 3:Chapters 2-5: despite the authors' broad knowledge of the presented methods, we do not have any references. If the reader wanted to delve deeper into the subject of a given method, he would not know where to look for the information that the authors provided.
Response 3:Thank you very much for your helpful comments. I have added the references. Along with that, we fixed the reference number. The reference number is written in red and shaded in yellow.
Comments 4:There is no information provided as to where the figures come from, whether they are from literature or private photos of the authors.
Response 4:Thank you very much for your helpful comments. All of them are our original photos and images.
Comments 5:Chapters 6-8: First 4 subsections describes the diseases, the last one is an explanation of the method. I don't really understand why the authors divided the section in this way. Moreover, in the flood of information it is hard to find a conclusion from the paper. The review should provide conclusions for the reader, I don't notice it in this work.
Response 5:Thank you very much for your important comments and your excellent suggestion. We have edited the sentence structure for making easy to understand. Moreover, we have added the conclusions in each section. (page4 line137-138, page6 line181-183, page11 line329-331, line339-342, page13 line398-402)
Comments 6:Chapter 9: again we don't know where the figure comes from. Comparisons of methods refer only to selected diseases, which is understandable, because otherwise it would go far beyond the scope of a single publication. However, the authors have described the previous chapters in great detail, which leaves some unsaid information. It is worth considering whether the work should not focus on a smaller scope, but described in a more precise manner. A positive aspect of the work is the tables, which make the analysis more readable, but they are missing in all comparisons.
Response 6:Thank you very much for your important comments and your excellent suggestion. Although we considered to explain in detail regarding to the treatment, as you mentioned, it is expected to become quite extensive. Because we believe that this journal focuses on the “diagnosis”, we have decided to refrain from discussing the treatment. Therefore, we have changed the title. (page1 line2-3)
Comments 7:Conclusions: does not contain any valuable conclusions that the reader would expect.
Response 7:Thank you very much for your important comments. We have added them. (page18 line530-536)
Reviewer 3 Report
Comments and Suggestions for Authors
Dear author
It was interesting to read your manuscript, which adds great value to the EUS investigation and suggests a wider utility of this procedure.
The manuscript is well-written and contains updates regarding the application of endoscopic ultrasound, especially in intraductal examination.
I suggest stating the type of review that you conducted – mainly a narrative review; this should be written in the manuscript because otherwise, it will be necessary to introduce the Methodology section and the research databases.
Moreover, it's essential to specify the source of the images used in your paper and whether the patients agreed to use them under GDPR and ethical regulations. This is a crucial ethical consideration in medical research.
Author Response
Dear Editors and Reviewers
Thank you very much for reviewing our manuscript and offering valuable advice.
We have addressed your comments with point-by-point responses, and revised the manuscript accordingly.
Responses to Reviewer 3
We appreciate the reviewer’s constructive comments.
Comments 1:I suggest stating the type of review that you conducted – mainly a narrative review; this should be written in the manuscript because otherwise, it will be necessary to introduce the Methodology section and the research databases.
Response 1:Thank you very much for your helpful recommendations. We believe this review is categorized as a narrative review. We have added it to the title of the main text. (page1 line3)
Comments 2:Moreover, it's essential to specify the source of the images used in your paper and whether the patients agreed to use them under GDPR and ethical regulations. This is a crucial ethical consideration in medical research.
Response 2:Thank you for your comment. We apologize our insufficient explanation regarding the source of the image. We used our original photos and images, and we have obtained prior consent from the patients before using the examination images for our research.
Round 2
Reviewer 2 Report
Comments and Suggestions for Authors
Thank the Authors for their contribution to making the work more readable for readers.
I have one issue about the methodology. It is not enough to add the method in the title. A paragraph should be added in which the authors describe what methods they used in their work, even if it is a narrative review.
Author Response
Dear Editors and Reviewers
Thank you very much for reviewing our manuscript and offering valuable advice.
We have addressed your comments with point-by-point responses, and revised the manuscript accordingly.
Responses to Reviewer 2
We appreciate the reviewer’s constructive comments.
Comments 1:I have one issue about the methodology. It is not enough to add the method in the title. A paragraph should be added in which the authors describe what methods they used in their work, even if it is a narrative review.
Response 1:Thank you for your kind suggestion. We have added the methodology. As a result, the section numbers have changed, and we have edited them accordingly. We have highlighted the changes in green.